# Ebola Virus Encodes Two microRNAs in Huh7-Infected Cells

**DOI:** 10.3390/ijms23095228

**Published:** 2022-05-07

**Authors:** Idrissa Diallo, Zeinab Husseini, Sara Guellal, Elodie Vion, Jeffrey Ho, Robert A. Kozak, Gary P. Kobinger, Patrick Provost

**Affiliations:** 1Centre Hospitalier Universitaire de Québec Research Center/CHUL Pavilion, Quebec, QC G1V 4G2, Canada; idrissa.diallo@crchudequebec.ulaval.ca (I.D.); zeinab.husseini@crchudequebec.ulaval.ca (Z.H.); sara.guellal@crchudequebec.ulaval.ca (S.G.); elodie.vion@crchudequebec.ulaval.ca (E.V.); jeffreyho1211@gmail.com (J.H.); 2Department of Microbiology, Infectious Diseases and Immunology, Faculty of Medicine, Université Laval, Quebec, QC G1V 4G2, Canada; 3Special Pathogens Program, National Microbiology Laboratory, Public Health Agency of Canada, Winnipeg, MB R3B 3M9, Canada; rob.kozak@sunnybrook.ca; 4Division of Microbiology, Department of Laboratory Medicine & Molecular Diagnostics, Sunnybrook Health Sciences Centre, Toronto, ON M4N 3M5, Canada; 5Galveston National Laboratory, Department of Microbiology & Immunology, University of Texas Medical Branch, Galveston, TX 77550, USA; gpkobinger@gmail.com

**Keywords:** RNA sequencing, microRNAs, Ebola virus, filoviruses, viral microRNA

## Abstract

MicroRNAs (miRNAs) are important gene regulatory molecules involved in a broad range of cellular activities. Although the existence and functions of miRNAs are clearly defined and well established in eukaryotes, this is not always the case for those of viral origin. Indeed, the existence of viral miRNAs is the subject of intense controversy, especially those of RNA viruses. Here, we characterized the miRNA transcriptome of cultured human liver cells infected or not with either of the two Ebola virus (EBOV) variants: Mayinga or Makona; or with Reston virus (RESTV). Bioinformatic analyses revealed the presence of two EBOV-encoded miRNAs, miR-MAY-251 and miR-MAK-403, originating from the EBOV Mayinga and Makona variants, respectively. From the miRDB database, miR-MAY-251 and miR-MAK-403 displayed on average more than 700 potential human host target candidates, 25% of which had a confidence score higher than 80%. By RT-qPCR and dual luciferase assays, we assessed the potential regulatory effect of these two EBOV miRNAs on selected host mRNA targets. Further analysis of Panther pathways unveiled that these two EBOV miRNAs, in addition to general regulatory functions, can potentially target genes involved in the hemorrhagic phenotype, regulation of viral replication and modulation of host immune defense.

## 1. Introduction

Ebola virus disease (EVD) caused over 28,000 infections and 11,000 deaths during the largest outbreak in West Africa (case fatality range: 26 to 67%) between 2013 and 2016 and remains a global public health threat [1]. During the period extending from 2018 to 2020, no less than 2300 deaths have been recorded in the eastern Democratic Republic of Congo (case fatality: 66%) due to EVD [1]. As evidenced by the COVID-19 crisis [2], the globalization of the world and climate change make us more vulnerable than ever to emerging diseases and the resurgence of epidemic upsurges, such as EVD [3].

Ebola virus (EBOV) is a negative sense RNA virus [4] that belongs to the *Ebolavirus* genus, which consists of six known species, including *Zaire ebolavirus* (to which EBOV belongs), *Sudan ebolavirus*, *Bundibugyo ebolavirus*, *Taï forest ebolavirus*, *Reston ebolavirus* and *Bombali ebolavirus* [5].

The molecular mechanisms of EBOV pathogenesis are being increasingly uncovered, but generally focus has been on the macromolecular aspect involving proteins. Indeed, the Ebola virus genome codes for nine proteins, including structural viral proteins VP24, VP30, VP35 and VP40, Nucleoprotein (NP), Glycoprotein (GP), soluble GP (sGP), small soluble GP (ssGP) and RNA-dependent RNA polymerase (L) [6]. Actors such as non-coding RNAs (ncRNAs), especially microRNAs (miRNAs), are poorly integrated in the pathophysiology of EVD.

miRNAs are short (19–24 nucleotides (nt)) non-coding RNA (ncRNA) sequences involved in fine-tuning the expression of ~60% of our genes. Most mammalian mRNAs are conserved targets of miRNAs, particularly in their 3’ untranslated regions (3′ UTRs) [7].

miRNAs are important players in viral pathogenesis, as viral infection affects host miRNA levels and the transcriptome as part of the host inflammatory response and may render the host more vulnerable [8,9]. In addition to its viral proteins, EBOV can presumably encode its own miRNAs to subvert host immune defenses [10,11,12].

Viral miRNAs (v-miRNAs) were first discovered in 2004 in Epstein–Barr virus, a DNA virus [13]. They were proven to play a role in the suppression of host immune response, regulation of viral replication and promotion of viral persistence [12,14,15]. The biogenesis of most DNA virus v-miRNAs is facilitated by host Drosha and Dicer machineries, similar to eukaryotic miRNAs [14]. Although v-miRNAs have been mostly attributed to DNA viruses, there is controversy surrounding the existence of v-miRNAs derived from RNA viruses mainly due to the lack of strong evidence [16].

Many researchers argue that RNA viruses do not express their own miRNAs due to the cytoplasmic localization of the virus during replication, the self-complementarity of the miRNA with the viral genome and the potential detrimental effect of miRNA processing on viral genome [17,18,19,20]. However, addressing the localization of the virus, there may exist noncanonical pathways for nucleus-independent miRNA processing [18,21] or there may be transient translocation of Drosha to the cytoplasm [18]. Additionally, Varble et al. [22] were able to successfully engineer an influenza virus to express miR-124, suggesting that RNA viruses are able to produce miRNA without an effect on their genomic fitness. There is also evidence supporting RNA viruses expressing their own miRNAs. Retroviruses, such as the human immunodeficiency virus 1 (HIV-1), a positive sense RNA virus, have been reported to express miRNAs that help regulate viral replication, and several miRNA candidates have been predicted for HIV-1 [23,24]. Moreover, our team and others have previously identified and characterized functional v-miRNAs released from the HIV-1 trans-activation response (TAR) element upon asymmetrical processing by Dicer [23,25,26]. These HIV-1-derived miRNAs were found to potentially regulate several candidate genes involved in apoptosis and cell survival, thereby creating a cellular environment more favorable to the replication and persistence of the virus [27]. Through small RNA deep sequencing analysis, functional miRNAs were also found in feline foamy virus [28].

Several studies have detected the presence of EBOV-derived miRNAs by analyzing small RNA sequencing (sRNA-Seq) data. Among them, Liu et al. [29] discovered an miR-155 analog, Zebov-1-5p, which presumably targets importin α5, a critical regulator in the interferon signaling pathways. Other studies described similar results and several v-miRNAs (miR-1-5p, miR-1-3p, miR-T3-3p) have been found in high abundance and predicted to target various genes involved in host cell defense, such as NF-kB and TNF, but the functionality of these v-miRNAs has not been experimentally validated [10,11,30,31]. Indeed, using RNA-Seq alone to detect EBOV-derived miRNAs is insufficient, as experimental validation of several computer-predicted EBOV-derived miRNAs has disproved them as bona fide miRNAs [32].

Recently, we have reported the first comparative miRNA transcriptome (miRNome) analysis of a human liver cell line (Huh7) infected with either of the two EBOV variants, Mayinga or Makona, or with *Reston ebolavirus* [33]. Following the characterization of the miRNome of cultured Huh7 infected with EBOV [33], we posited that the virus may also produce miRNAs which could potentially interfere with translation of specific human host messenger RNAs (mRNAs) and contribute to EBOV pathogenesis. We discovered one potential EBOV-derived miRNA from EBOV Mayinga-infected cells and another from EBOV Makona-infected cells. Using miRDB [34], these miRNAs have been predicted to target genes that are involved in promoting viral replication, modulating hemorrhagic phenotype and suppressing antiviral immune responses.

## 2. Results

### 2.1. Description of the Two EBOV-Encoded miRNA Candidates

We recently documented differentially expressed miRNAs in cultured human liver cells upon infection with EBOV [33] and we sought to verify whether v-miRNAs were produced upon EBOV infection. Following this sRNA-Seq study, we used the miRDeep2 algorithm [35,36] to identify potential EBOV-derived miRNAs. Considering that EBOV miRNAs may have different properties to host miRNAs, we used less stringent settings (without any threshold set on the scores). We identified two v-miRNAs from two variants of EBOV (Figure 1): Mayinga (GenBank Sequence Accession: NC002549 (Genome Locus chrNC002549:113..158:-)) and Makona (GenBank Sequence Accession: KJ660347 (Genome Locus chrKJ660347:11497..11575:-)). No v-miRNAs were predicted from the *Reston ebolavirus* genome (GenBank Sequence Accession: NC_004161), which was included in our analyses. *Bundibugyo ebolavirus* (BDBV), *Taï Forest ebolavirus* (TAFV), *Bombali ebolavirus* (BOMV) and Marburg virus (MARV) were not included in the miRDeep2 analysis but appeared to encode the sequences of the two identified viral miRNAs, with a coverage ranging from 50 to 100% depending on the isolate (data not shown, BLASTN, NCBI).

The two putative miRNAs are tentatively named EBOV-miR-MAK-403 and EBOV-miR-MAY-251 in reference to their respective origins. The estimated probabilities that miR-MAK-403 and miR-MAY-251 candidates are positive were 64 (+/−48%) and 57% (+/−50%), respectively. The secondary structures of their precursors exhibited the stem–bulge–loop typical of classical RNase III family (like Dicer) substrates (Figure 1). Looking at the loci from which miR-MAY-251 and miR-MAK-403 might derive, we found that miR-MAY-251 has a perfectly complementary region located upstream of the coding sequence of the nucleoprotein (NP) gene mRNA. On the other hand, the miR-MAK-403 sequence has a region with partial complementarity (with a single mismatch) located upstream of the coding sequence of the viral RNA-dependent RNA polymerase (L) gene mRNA (Figure 2).

sRNA-Seq results showed that both miRNAs were almost absent at 24 h and accumulated only in the late phase of infection (Figure 3A). We also confirmed these observations by RT-qPCR (Figure 3B). When compared in terms of relative transcripts per million (TPM; data not shown), miR-MAY-251 appears to be twice as abundant as miR-MAK-403, i.e., 2.82 × 10^5^ for the former and 1.36 × 10^5^ for the latter. Both miRNAs appear to be specific to their respective genomes, as miR-MAY-251 was not detected upon infection with EBOV Makona and miR-MAK-403 was absent from EBOV Mayinga-infected cells (Appendix A). Using sRNA-Seq, bioinformatics tools and RT-qPCR, we report the existence of two novel v-miRNAs whose biogenesis and functions remain to be determined.

### 2.2. miR-MAY-251 and miR-MAK-403 Potential In Silico Human Host Target Candidates

To study the potential functions of the two miRNAs in the context of human cell infection, we first subjected them to miRDB [34] target prediction. There were 881 and 665 predicted targets for the two 22 nt long EBOV miR-MAY-251 and miR-MAK-403 (see Appendix A); 23 to 27% of the targets had scores of more than 80%, with higher scores representing higher statistical confidence in the prediction result.

We next sought to predict the putative overall function of these two v-miRNAs in infected cells. Using PANTHER classification algorithms [38], we analyzed the three aspects of GO (molecular function, biological process, cellular component) of those target genes with a score higher than 80%. The results are summarized in Figure 4.

The molecular-level activities performed by miR-MAK-403 and miR-MAY-251 are predicted to target gene products involved in binding (GO:0005488), catalytic activity (GO:0003824) and molecular function regulation (GO:00988772). The major GO terms for the biological process aspect were related to: biological regulation (GO:0065007), cellular process (GO:0009987) and metabolic process (GO:0008152). The targets of the two v-miRNAs perform their functions principally in: cellular anatomical entity (GO:0110165), intracellular (GO:0005622) and protein-containing complex (GO:0032991) (Figure 4A,B). Although some variations exist in the ratios, the targets of the two miRNAs also share similarities in their protein class and PANTHER pathway (Appendix A). The targets belong to classes, such as protein-modifying enzyme (PC00260), nucleic acid metabolism (PC00171), scaffold protein (PC00226), transporter (PC00227), etc. Concomitantly, PANTHER pathway enrichment revealed the involvement of miR-MAK-403 target genes in angiogenesis (P00005), the epidermal growth factor (EGF) signaling pathway (P00018) and blood coagulation (P00011), in addition to many other signaling pathways, such as platelet-derived growth factor (PDGF) pathways. Similarly, the majority of miR-MAY-251-associated target genes were also enriched in pathways involved in cell–cell interaction (cadherin and WNT pathways, P00012, P00057), angiogenesis and many others (Appendix A).

The above functional predictions suggest that the two EBOV miRNAs, miR-MAY-251 and miR-MAK-403, might exert their regulatory effects on these potential targets and modulate specific cellular functions in a way that might contribute to virus pathogenicity.

### 2.3. EBOV miR-MAY-251 May Modulate Selected Host mRNA Targets

In an attempt to experimentally validate the above in silico predictions, we employed RT-qPCR to assess the potential regulatory effect of EBOV miRNAs on their predicted targets. To do this, Huh7 cells were transfected with synthetic miR-MAY-251 RNA mimics or with a synthetic miRNA mock (control), after which we assessed the mRNA levels of six selected miR-MAY-251 in silico-predicted host targets, selected based on their energy interactions with the EBOV miRNAs and their links with viruses and the immune system reported in the literature (Appendix A): dual specificity protein phosphatase 16 (DUSP16), Nicotinamide phosphoribosyltransferase (NAMPT), Pumilio RNA Binding Family Member 2 (PUM2), FYVE, RhoGEF and PH domain-containing protein 1 (FDG1), Netrin receptor (UNC5D), and phospholipase C beta 4 (PLCB4). Data indicated that miR-MAY-251 transfection had no significant effect on the level of NAMPT, PUM2, FDG1 or PLCB4 transcripts (Figure 5). Nonetheless, though contrariwise, DUSP16 and UNC5D transcript levels were significantly modulated upon EBOV miRNA mimic transfection. Compared to the control, DUSP16 levels decreased by about 15% and 65% upon transfection with 100 nM and 200 nM of miR-MAY-251 mimic, respectively. As for UNC5D mRNA, it showed a significant upregulation, reaching 28× higher levels at 200 nM mimic concentration compared to the control (Figure 5). The above data support the functionality of EBOV miR-MAY-251 and the possibility of a gene regulatory effect on its in silico-predicted host mRNA targets.

### 2.4. EBOV miR-MAK-403 May Modulate Selected Host mRNA Targets

We next used the same transfection and RT-qPCR settings to validate human host mRNA targets of EBOV miR-MAK-403. The mRNA transcript level of the following eight selected host targets of miR-MAK-403 was quantitated upon mimic transfection (Appendix A): cyclin-dependent kinase 13 (CDK13), E3 ubiquitin ligase (SMURF2), WD repeat-containing protein 7 (WDR7), P2Y purinoceptor 13 (P2RY13), VAMP associated protein A (VAPA), phosphatidylinositol polyphosphate 5-phosphatase type IV (INPP5E4), cohesin subunit SA-2 (STAG2) and cell growth regulator with ring finger domain 1 (CGRRF1).

We observed significant downregulation of mRNA transcript levels of four targets (CDK13, SMURF2, WDR7 and P2RY13) compared to the control (Figure 6). This downregulation was reported at both mimic concentrations and seemed to be dose-dependent. Conversely, STAG2 demonstrated significant upregulation compared to the control. Additionally, three mRNA targets (INPP5E4, CGRRF1, VAPA) were upregulated at one or both mimic concentrations, but this was not statistically significant when compared to the control (Figure 6).

Altogether, these results suggest a potential modulatory effect of EBOV miR-MAY-251 and miR-MAK-403 on their in silico-predicted host mRNA targets.

### 2.5. miR-MAY-251 and miR-MAK-403 May Directly Regulate DUPS16 and CDK13 through 3′ UTRs, Respectively

Human host DUSP16 and CDK13 genes were chosen for additional downstream analyses. In addition to their pronounced and significant modulation observed, DUSP16 and CDK13 have been described in the literature as being potentially involved in viral replication [39,40] and in the regulation of a number of cellular processes via kinase and cytokine pathways [36,37,38], respectively.

miRNAs exert their functions through direct interaction with the target mRNA 3′ UTR. With perfect or partial complementarity, the seed regions of miRNAs bind to the 3′ UTR of their cognate mRNAs, leading to translation repression or initiation of mRNA decay [7]. Eventually, such miRNA–mRNA interactions lead to downregulation of the target at the transcript or protein levels. Since both DUSP16 and CDK13, compared to other tested targets, were most significantly downregulated by miR-MAY-251 and miR-MAK-403, respectively, we sought to study the potential interaction between these two mRNAs and their corresponding miRNAs. Using dual luciferase reporter assays, we assessed whether miR-MAY-251 and miR-MAK-403 could regulate the expression of a luciferase reporter gene whose 3′ UTR included a segment of the 3′ UTR of either DUSP16 or CDK13 mRNA comprising two or more binding sites for miR-MAY-251 and miR-MAK-403, respectively (plasmids with WT 3′ UTRs). Plasmids with 3′ UTRs in which the binding sites for the two EBOV miRNAs were mutated were used as negative controls (plasmids with MUT 3′ UTR).

Huh7 cells were co-transfected with DUSP16 or CDK13 plasmids and miR-MAY-251 or miR-MAK-403 mimics at different concentrations (0, 50, 100 and 200 nM) or with a synthetic RNA control (mock; Figure 7). Compared to the mock control, miR-MAY-251 transfection significantly decreased (33–42%) luciferase expression when placed under DUSP16 mRNA 3′ UTR. This inhibitory effect was relieved upon mutation of the binding sites for EBOV miR-MAY-251 in the DUSP16 mRNA 3′ UTR (Figure 7, upper panel), thereby validating their functionality and possible implication in the downregulation of DUSP16 mRNA levels seen in EBOV Mayinga-infected Huh7 cells.

The regulatory effects of EBOV miR-MAK-403 on luciferase expression placed under the CDK13 mRNA 3′ UTR was less evident, as inactivation of the putative miR-MAK-403 binding sites in the CDK13 mRNA 3′ UTR had no significant effect (Figure 7, lower panel), hence the importance of experimentally validating v-miRNA–host mRNA target predictions.

## 3. Discussion

While various DNA viruses have been documented to encode miRNAs [13], the ability of RNA viruses to produce miRNAs has long been controversial and the debate over the authenticity of the previously reported RNA virus-produced miRNAs remains. Nevertheless, a few studies have described the identification and functional characterization of a number of RNA virus-encoded miRNAs. The majority of these v-miRNAs belong to RNA viruses of the Retroviridae family (retroviruses), whose replication cycle includes a nuclear step upon the reverse transcription and integration of their genome into the host [16,41]. HIV-1 and bovine leukemia virus (BLV) retroviruses were reported to produce at least one miRNA and these miRNAs are speculated to affect viral replication or host biology [16,26]. However, the evidence of miRNA being produced by non-retroviral RNA viruses is limited to a few examples, such as the human pathogen EBOV and the silkworm pathogen Bombyx mori cypovirus (BmCPV) [10,29,31,42].

The present study identified two miRNAs, miR-MAY-251 and miR-MAK-403, encoded by the Mayinga and Makona EBOV variants, respectively. In contrast, no v-miRNA was detected in Huh7 cells infected with RESTV. Nevertheless, in addition to EBOV, the sites of both miR-MAY-251 and miR-MAK-403 appeared to be relatively well evolutionary conserved across all sequenced filoviruses, such as RESTV, TAFV, BDBV and MARV, depending on the isolates. Further studies are warranted to understand whether they are able to produce miR-MAY-251 and miR-MAK-403 upon infection as well as the conditions required for such production to occur.

Validated using RT-qPCR, the two EBOV miRNAs are 22 nt in length, mirroring the typical size of known miRNAs (19–24 nt). The fact that each of the two novel v-miRNAs was detected only in cells infected with one of the EBOV variants but not the other, supports the notion that these are not mere contaminants or degradation products introduced during RNA extraction or even RNA library preparation. Comparison of the v-miRNA expression profile with that of the host cell revealed that miR-MAY-251 and miR-MAK-403 are, respectively, about nine-fold and six-fold more abundant than miR-122, one of the most abundant miRNAs found in Huh7 cells [33]. Compared to miR-148a, another highly abundant miRNA in Huh7 cells, miR-MAY-251 and miR-MAK-403 were 3 and 1.5 times more abundant, respectively. The abundance of both v-miRNAs supports their active production and functional potential. We relied mainly on in silico tools to predict the nature of these two v-miRNAs. Secondary structure predictions by the miRDeep2 RNAfold tool indicated that the precursor structures (pre-miRNAs) of miR-MAY-251 and miR-MAK-403 resemble the typical miRNA hairpin structure required for proper recognition by Dicer for further miRNA processing [43]. Though miRNA precursors are not the only RNA molecules capable of forming secondary structures, as most of the other noncoding RNAs display folding properties [44], possessing a lower number of bulges of reduced size (number of bulges: ∼1.86 bulges; defined bulge size: ∼1.47 nt) might qualify an RNA substrate for RNase III processing and classification as an miRNA precursor. In our case, structure predictions of miR-MAY-251 and miR-MAK-403 precursors present no more than two bulges and/or internal loops.

Although little evidence supports the existence of RNA virus-encoded miRNAs, the production of small non-coding RNAs which might not fall into the miRNA category is more widely accepted in the field [32]. Indeed, drawing a strong conclusion about the nature of a newly identified RNA necessitates the employment of many experiments investigating whether the biogenesis and/or mode of action of the RNA in question mirrors the minimal criteria reported for other members of the family to which they are to be classified into. Hence, there is a need for further independent validation of our data.

Our sRNA-Seq data are unambiguous about the existence of these two v-miRNAs, which was confirmed by qPCR. Moreover, using the search keywords “blood”, “ebola”, “rna-seq”, ‘’transcriptome” and “miRNome”, we examined RNA transcripts in the NCBI Sequence Read Archive for miR-MAY-251 and miR-MAK-403 in the blood of people infected with EBOV during the different outbreaks (see Appendix A). Intriguingly, only miR-MAY-251 was detected in the biobanks, in the bioproject PRJNA23122.1 (submitted 11 December 2013) containing 3657 Sequence Read Archive (SRA) experiments. These data were collected by three major institutions: the U.S. Army Medical Research Institute for Infectious Diseases (USAMRIID), Public Health England (PHE) and the University of Texas Medical Branch (UTMB). miR-MAY-251 was found in the blood of patients infected with the Mayinga (1976, isolate R3816) and Kikwit (1995, isolate 9510621; R4317a) variants, two genetically close variants, both discovered in the Democratic Republic of Congo (see Appendix A). A search among these samples’ data yielded between 9 and 2110 matches (average: 723) for miR-MAY-251. In our RNA-Seq data, miR-MAK-403 was two times less abundant than miR-MAY-251, which could explain its escape from detection in the blood of people infected with the Makona variant.

miR-MAY-251 and miR-MAK-403 appear to derive from the NP and L genes, respectively (see Figure 2). Therefore, they presumably derive from the negative sense RNA strand rather than from the positive sense RNA strand or from viral mRNAs.

The qPCR analysis of the viral NP and GP transcripts (we did not monitor the L gene) showed that they were highly expressed at the early stage of infection (24 h) and more abundant in the later stage of infection (96 h). The fact that EBOV miRNA levels were significantly higher only at the late stage of viral infection is temporally consistent with their biosynthesis from EBOV transcripts and suggests their possible involvement in modulating specific viral and/or host functions at later, rather than earlier, stages of the virus life cycle. This is consistent with what has been reported for other viral miRNAs, specifically those which go through a latency period, during which they produce miRNAs to help them evade the host immune system or limit their own replication [26,45].

The processing of viral mRNA to produce v-miRNAs was documented in HIV-1, where an miRNA is produced from a hairpin structure in a Dicer-mediated process [26]. The fact that these v-miRNAs may derive from the negative sense viral RNA strand constitutes one of the reasons why the existence of RNA virus-derived miRNAs is not widely accepted in the scientific community. It is believed that the production of v-miRNAs in such a way would greatly impair the viral genome as it would involve viral genome cleavage and because of the potential antigenome effects these miRNAs may exert [46]. Nonetheless, processing of the viral RNA genome may release a v-miRNA, which, in turn, may contribute to regulating critical aspects of the virus life cycle and be advantageous to the virus. The notion that both v-miRNAs derive from elements located upstream of the coding sequences of two viral genes might imply one way through which EBOV might regulate its own genome expression/replication in host cells, as seen in other viruses employing v-miRNAs to control their own replication [47].

Further studies are warranted to confirm the legitimate identity of these v-miRNAs and to investigate their function, relevance and importance in EBOV replication and pathogenesis. EBOV-derived miRNAs would be expected to associate with an Argonaute protein of the RISC complex to mediate their function. Regarding miR-MAY-251 and miR-MAK-403 biogenesis, Drosha and/or Dicer may not be necessarily involved, as reports have described several non-canonical (Drosha- or Dicer-independent) pathways for miRNA production [16,48]. BLV, for example, generates polymerase III-derived sub-genomic transcripts independent from Drosha, from which v-miRNAs are derived and affect the nuclear–cytoplasmic localization of the host miRNome [15]. A number of commonly known eukaryotic miRNAs bypass some of the steps of canonical miRNA biogenesis and are produced in a Drosha- and/or Dicer-independent manner [48]. Such non-canonical pathways tend to favor the production of 5p miRNAs, which is intriguing considering that the 5p forms of miR-MAY-251 and miR-MAK-403 were much more abundant than their 3p counterparts (Figure 1). Several functions have been attributed to virally encoded miRNAs, including modulation of host processes, mainly immune responses, and regulation of viral replication and gene expression pathways [12]. For instance, EBOV miR-1-5p was reported to target importin-α5, inducing deregulation in the INF pathway and contributing to virus pathogenicity in patients [29]. In an attempt to uncover the potential functions of miR-MAY-251 and miR-MAK-403 in virus pathogenesis, we used computational prediction followed by experimental validation, as several studies have previously employed to characterize v-miRNAs from numerous viruses [12,13]. The PANTHER database helped us to cluster the two putative EBOV miRNA targets into GO groups based on their biological processes, molecular functions, and cellular compartments. As illustrated in Figure 2 and Appendix A, multiple gene pathways were enriched in our analyses, such as the protein-modifying enzyme pathway (PC00260). Several studies demonstrated that EBOV transcription may modulate host protein-modifying enzymes. For instance, both Protein phosphatase 1 (PP1), which dephosphorylates VP30 protein [49], and Serine-arginine protein kinase 1 (SRPK1) [50] are required to support primary viral transcription as well as the re-initiation of VP30-mediated transcription at internal EBOV genes [50]. Other protein-modifying enzymes were previously described as regulators of EBOV transcription, such as the host ubiquitin ligase RBBP6, whose knockdown stimulated viral transcription and increased EBOV replication [51]. These processes are tightly regulated by EBOV components, among which v-miRNAs may contribute to preventing the host machinery from interfering with viral transcription.

We also observed that the EGF signaling pathway may be modulated by miR-MAK-403, which is consistent with a study demonstrating that inhibitors of EGF receptors (EGFRs) blocked filoviral GP-mediated entry and prevented growth of replicative EBOV in Vero cells [52]; EGFR may thus facilitate EBOV replication. The cadherin pathway is among those enriched for miR-MAY-251, which may contribute to virus-induced changes in the protein organization of the endothelial–cell junctions, in particular, the cadherin–catenin complex of the vascular endothelium. This might be one explanation of the imbalance of fluid between the intravascular and extravascular tissue spaces that is usually manifested as oedema among EVD patients [53].

Recently, we demonstrated in the hepatic cell line Huh7 that EBOV Mayinga and Makona hemorrhagic phenotypes may result from an imbalance in coagulation factor levels [33]. Incidentally, PANTHER analysis revealed that miR-MAK-403 target genes may be involved in coagulation disorder and fibrinolysis, two critical parameters in EBOV pathogenesis [54]. By targeting the plasminogen activation cascade (P00050), which is the primary catalyst of fibrin degradation [55], angiogenesis (P00005) [56], the vascular endothelial growth factor (VEGF) signaling pathway (P00056) [57], the PDGF signaling pathway (P00047) [58] and the TGF-β signaling pathway (P00052) [59], miR-MAK-403 could lead to coagulation disorder and exacerbate hemorrhaging while impairing host defenses. This is also a possible scenario for miR-MAY-251, since it targets genes involved in integrin signaling (P00034) and inflammation mediated by chemokine and cytokine signaling pathways (P00031, PANTHER pathway data; Appendix A). In fact, integrin signaling controls adhesion and aggregation at the site of cell injury [60] and is involved in the interplay between inflammation and coagulation [61]. The richness of coagulation-related terms (PANTHER pathway data) among the genes targeted by EBOV-derived miRNAs deserves further investigation, as it may be the key to this still poorly understood EVD phenotype.

Among the potential targets of miR-MAK-403 and miR-MAY-251 identified in silico, the ones with the highest prediction scores in the miRDB were tested (Appendix A). Many of them, such as WDR7 [62], INPP5E [63], CDK13 [39,40], VAPA [64,65,66], NAMPT [67,68,69] and PUM2 [70], are known to be involved in viral infection mechanisms, whether at the point of the viral entry into the cell or during viral replication. v-miRNA targets, such as P2RY13 [71,72], SMURF2 [73,74], CGRRF1 [75], STAG2 [76], DUSP16 [77,78], FGD1 [79], UNC5D [80] and PLCB4 [81], may also be involved in the cellular response to infection and cellular viability. We were unable to document a significant change in the expression of INPPE5, CGRRF1 and VAPA genes upon miR-MAK-403 transfection, or in the expression of NAMPT, PUM2, FGD1 and PLCB4 genes upon miR-MAY-251 transfection, suggesting that these genes be considered as false positives, which are often seen with bioinformatics tools [82].

However, we observed a significant increase in UNC5D and STAG2 expression after the respective transfection with miR-MAY-251 and miR-MAK-403 mimics. Acting canonically, miRNAs negatively regulate their mRNA targets. However, miRNAs may counterintuitively lead to an upregulation of mRNA targets [83,84], either through direct interaction with the mRNA target [85] or indirectly through increasing mRNA stability and translation rates [86]. The upregulated UNC5D target is involved in the inhibition of cellular migration and proliferation [87]. It is also involved in apoptosis regulation, where, in the presence of Netrin-1, UNC5D can transduce a survival signal and inhibit apoptosis [88,89]. Analogously, we previously showed that EBOV induced an upregulation of anti-apoptotic Bcl-2 gene at the early stage of the EBOV Mayinga or Makona infection [33]. It would be interesting to characterize this pathway in EBOV-infected Huh7 cells, especially Netrin-1, to ascertain whether, as in Hepatitis C virus infection [90], the level and translation of Netrin-1 are likewise augmented in the context of EBOV infection. Upregulated upon miR-MAK-403 mimic transfection, STAG2 participates in viral replication as a transcription co-activator [91]. This suggests that EBOV may stimulate the transcription of the host STAG2 gene through its v-miRNA to improve replication of its own genome.

The only studied target gene whose expression was significantly downregulated after transfection with the miR-MAY-251 mimic was DUSP16, which is involved in cell viability and migration in hepatocellular carcinoma cell lines [77], and possibly improves viral replication.

Expression of P2RY13, an interferon-stimulated gene, is significantly downregulated after transfection with miR-MAK-403. P2RY13 protects the host against viral infections by releasing its ligand, ADP, as a danger signal during viral infection, hence limiting the replication of both DNA and RNA viruses [71]. CDK13, which was markedly downregulated by miR-MAK-403, was reported to restrict HIV-1 viral replication [40], whereas its silencing leads to a significant increase in virus production [40]. Together, these findings suggest a strong positive regulation of EBOV replication through miR-MAK-403.

Another target which was inhibited upon miR-MAK-403 transfection is SMURF2, a negative regulator of virus-induced IFN-β response [73]. SMURF2 interacts directly with EBOV VP40 matrix protein, which is essential for viral assembly and budding from the host cell [92], to improve viral budding [93]. SMURF2 is also a negative regulator of TGF-β signaling, which gives it an anti-apoptotic activity and suggests that it may promote cell migration [74,94,95,96]. Thus, we posit that the pro-viral activity of miR-MAK-403 would be exerted through inhibition of cell migration rather than through viral budding. Regulation of cell migration seems to be common to the EBOV Mayinga and Makona variants, each acting through their own v-miRNAs.

Reporter gene activity assays employing partial fragments of DUSP16 and CDK13 3′ UTRs confirmed their potential regulation by the EBOV v-miRNAs miR-MAY-251 and miR-MAK-403, respectively. The introduction of the entire 3′ UTR into the reporter gene construct would have enabled a better evaluation of what is occurring in vivo, including possible folding structures exposing or hiding certain binding sites. However, the length of their 3′ UTRs exceeded 3 kb, which made its cloning challenging. We, therefore, opted to use 3′ UTR fragments (500–700 bp) comprising the v-miRNA binding sites predicted by Targetscan. The lack of significant reporter gene modulation by v-miRNAs in that setting (i) may be related to the presence of endogenous regulatory miRNAs, as indicated by the decrease in reporter gene activity before transfection (0 nM), and (ii) may not preclude an effect in vivo, since RNAhybrid predicts that both DUSP16 and CDK13 mRNAs have additional potential binding sites for v-miRNAs in their 3′ UTRs as well as in their 5′ UTRs or coding regions.

Uncovering the existence of EBOV-derived miRNAs challenges the belief that RNA viruses do not express their own miRNAs, provides new insights into ebolavirus pathogenesis and calls for further independent investigations of their role, function and importance in EBOV replication.

## 4. Materials and Methods

### 4.1. Viruses

Experiments involving the manipulation of Ebolavirus were conducted with all precautions in the Biosafety Level 4 facility of the National Microbiology Laboratory of Canada (Winnipeg, MB). This study utilized the following EBOV isolates: EBOV/May (Ebola virus/H.sapiens-tc/COD/1976/Yambuku-Mayinga; NC_002549.1) and EBOV/Mak-C07 (Ebola virus/H. sapiens-tc/GIN/2014/Makona-WPGC07; KJ660347.2). Huh7 cells were infected with the variants separately at a multiplicity of infection (MOI) of 1.0. Cells were harvested at early (24 h) and late (96 h) stages of infection in biological triplicate (n = 3). Viral replication was assessed by RT-qPCR detection of GP mRNA, as described previously [33].

### 4.2. Cell Culture Conditions and Transfections

The hepatocyte-derived cellular carcinoma cell line Huh7 (a gift from Dr. Ralf Bartenschlager) was grown in Dulbecco’s modified Eagle’s medium (DMEM, Wisent, St-Bruno, Canada, cat. no. 219-010-XK) supplemented with 10% fetal bovine serum, 1 mM L-glutamine, 100 units/mL penicillin and 100 µg/mL streptomycin. Cells were grown and maintained in tissue culture plates and incubated at 37 °C in a humidified atmosphere containing 5% CO_2_. Cells were kept in exponential growth phase and subcultured every 2–3 days.

### 4.3. Plasmid Constructs

The wild-type (WT) sequences of DUSP16 3′ UTR (NCBI acc. no. NM_030640.3) and CDK13 3′ UTR (NCBI acc. no. NM_003718.5) were designed using gBlocks^®^ gene fragments (Integrated DNA Technologies, Inc., Coralville, IA, USA). Given the length of their 3′ UTRs (3551 and 4000 nt, respectively) and the difficulty of cloning such long fragments, we opted for shorter segments which carried potential binding sites for EBOV-derived miRNAs. Thus, 727 nt of the DUSP16 3′ UTR and 403 nt of the CDK13 3′ UTR were introduced downstream of the *Renilla* luciferase (Rluc) reporter gene in the XhoI/NotI cloning sites of the psiCHECK2 vector (Promega, Madison, WI, USA). Reporter constructs bearing the mutated versions of these 3′ UTRs were also engineered. The details of the construction strategy are summarized in Appendix A. All the constructs were independently confirmed by DNA sequencing at the Plateforme de Séquençage et Génotypage des Génomes (Centre de Recherche du CHU de Québec—CHUL, QC, Canada).

### 4.4. Cell Transfection and Dual Luciferase Assay

Three hundred thousand (300,000) Huh7 cells were cultured in 6-well plates and transfected the following day at 70–80% confluency using polyethylenimine [97] (PEI; Sigma, ON, Canada, cat. no. 919012) or lipofectamine 2000 [98] (Invitrogen, ON, Canada, cat. no. 11668019), as described previously, with slight modifications [33]. The EBOV miRNA mimics and/or plasmids to be transfected, as well as the transfection reagents, were diluted in Opti-MEM^®^ (Invitrogen, Burlington, ON, Canada, cat. no. 31985062). Forty-eight (48) h after transfection, cells were washed with PBS and lysed with 500 μL of the passive lysis buffer. Luciferase activities were measured using the Dual-Luciferase^®^ Reporter Assay System (Promega, Madison, WI, USA, cat. no. E1980) on a luminometer (TECAN INFINITE M1000 PRO, Tecan Austria GmbH, Grödig, Austria), according to the manufacturer’s instructions. Rluc activity was expressed relative to expression of the internal control *Firefly* luciferase (Fluc). Rluc expression was further normalized to the control in which cells were co-transfected with synthetic unrelated miRNA mimic, elsewhere referred to as mock control. All assays were conducted in triplicate in a 96-well format.

### 4.5. RNA Isolation

Total RNA was extracted from Huh7 cells using TRIzol reagent (Invitrogen, Burlington, ON, Canada, cat. no. 15596026) following the manufacturer’s recommendations, as described previously [33]. All RNA samples were subjected to treatment with DNAse I (TURBO DNA-free Kit, Invitrogen, Burlington, ON, Canada, cat. no. AM1907), analyzed quantitatively with a NanoDrop™ 2000 Spectrophotometer (Thermo Scientific™, cat. no. ND-2000, Waltham, MA, USA) and kept at −80 °C for subsequent analyses.

### 4.6. RT-qPCR

Detection of host mRNA targets. Total RNA (1 µg) extracted from host infected cells was reverse transcribed into cDNA with the HiFlex miScript II RT Kit (Qiagen, Germantown, MD, USA, cat. no. 218160), following the manufacturer’s protocol, as described previously [33]. After a 1/10 dilution of the cDNA, qPCR was performed using the SSo Advanced SYBR Green mix (Bio-Rad, Hercules, CA, USA, cat. no. 1725271) in a 0.1 mL MicroAmp™ Fast Optical 96-Well Reaction Plate (Applied Biosystem™,cat. no. 4346907, Burlington, ON, Canada) or Multiplate^®^ PCR plate™ (Bio-Rad, Hercules, CA, USA, cat. no. MLL9601). The final concentration of the primers (Integrated DNA Technologies, Inc.) used in RT-qPCR was 500 nM and their sequences are listed in Appendix A. The primers were designed with the Primer-BLAST tool [99]. Primers were chosen to allow specific amplification of the target mRNAs (span exon–exon junction). Temperature gradient tests were performed to determine the best annealing temperatures for each primer pair. Unless otherwise specified, qPCRs were performed on a StepOne™ Real-Time PCR System (Thermo Scientific™, cat. no. 4376357) or a CFX Connect Real-Time PCR Detection System (Bio-Rad, cat. no. 1855200). Unless otherwise specified, all data obtained (with StepOne™ Software, v2.3, Thermo Fisher Scientific Inc., Mississauga, ON, Canada) were normalized with reference genes and reported to the controls. The relative quantitation was calculated using the ∆∆Ct method [100].

Detection of viral miRNAs in infected cells. Total RNA (1 µg) extracted from host infected cells was reverse transcribed into cDNA with the miRCURY LNA RT Kit (Qiagen, cat. no. 339340). Custom miRNA primer sets designed by Exiqon^®^ (Vedbæk, Denmark) experts and optimized with LNA technology [101] were used (Qiagen). More precisely, we used the miRCURY LNA miRNA Custom PCR Assay (Qiagen, Germantown, MD, USA, cat. no. 339317), which contains forward and reverse primers for 200 reactions (cat. no. YCP0361677, for EBOV-miRNA-MAK-403-KJ660347_1 primers; cat. no. YCP0361680, for EBOV-miRNA-MAY-251-NC002549 primers). After a 1/10 dilution of the cDNA, qPCR was performed using the miRCURY LNA SYBR Green PCR Kit (Qiagen, cat. no. 339346) with a CFX Connect Real-Time PCR Detection System (Bio-Rad, cat. no. 1855200) in 96-well plates (Multiplate^®^ PCR plate™; Bio-Rad, cat. no. MLL9601), following the manufacturer’s protocol. The UniSp6 RNA spike-in was used for cDNA synthesis and PCR amplification normalization.

### 4.7. Statical Analysis of qPCR Data

The statistical method used in each case is mentioned in the figure captions. All statistical analyses were performed using GraphPad Prism version 9.3.1 (GraphPad Software, Inc., La Jolla, CA, USA), with statistical significance set at *p* < 0.05.

### 4.8. Illumina Nextseq Sequencing of Cells Infected with EBOV

All the data related to the sequencing and the methodology of their acquisition have been published previously (see reference [33]).

## Figures and Tables

**Figure 1 ijms-23-05228-f001:**
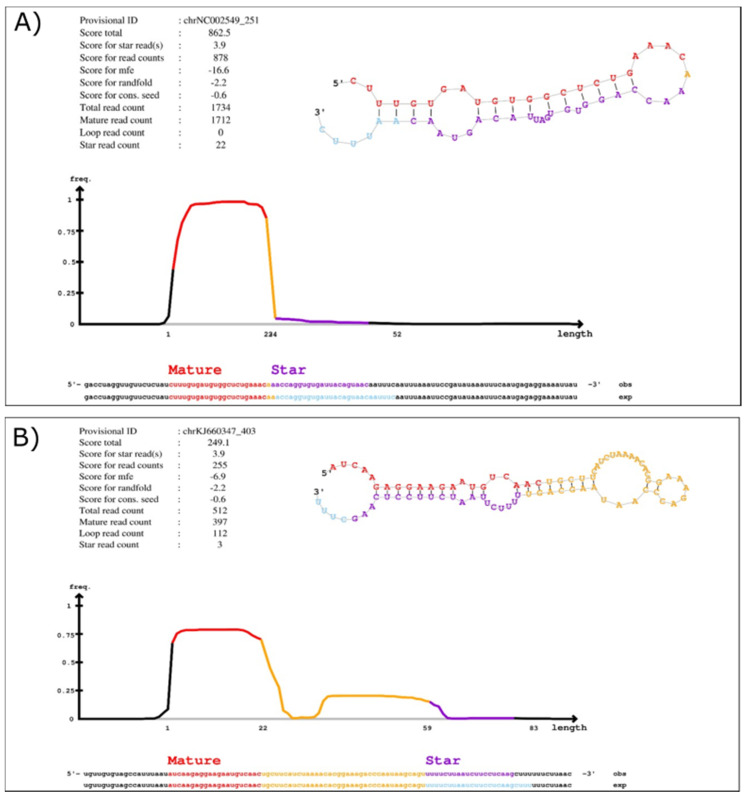
sRNA-Seq and miRDeep2 analysis of Huh7 cells infected with EBOV unveiled two novel EBOV microRNAs. EBOV-miR-MAY-251 (**A**) and EBOV-miR-MAK-403 (**B**) microRNAs were detected in Huh7 cells infected with Mayinga or Makona EBOV variants, respectively. In each panel, the upper left table lists the miRDeep2 score breakdown for the reported microRNAs, along with the read counts for the mature, loop and star sequences [37]. At the upper right of each panel are illustrations of the microRNA precursor secondary structure predictions. The bottom density plot highlights the read distribution in the predicted precursor sequence. The position of the star strand, as expected from Drosha/Dicer processing, is shown in light blue, while the star consensus position, as detected in the sequencing data, is shown in purple [37]. The complete data for the alignment are in Appendix A.

**Figure 2 ijms-23-05228-f002:**
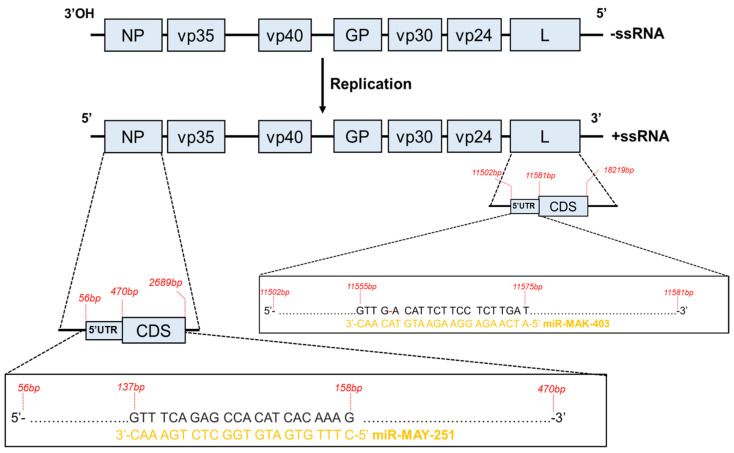
Schematic representation of the EBOV genome, transcribed mRNAs and two potential microRNAs. The two EBOV-derived microRNAs, miR-MAY-251 and miR-MAK-403, are located in the 5′ untranslated region (5′ UTR) of NP (137–158 bp) and L (11555–11575 bp), respectively. Negative (−)/Positive (+); ssRNA, single-strand RNA; CDS, coding DNA sequence.

**Figure 3 ijms-23-05228-f003:**
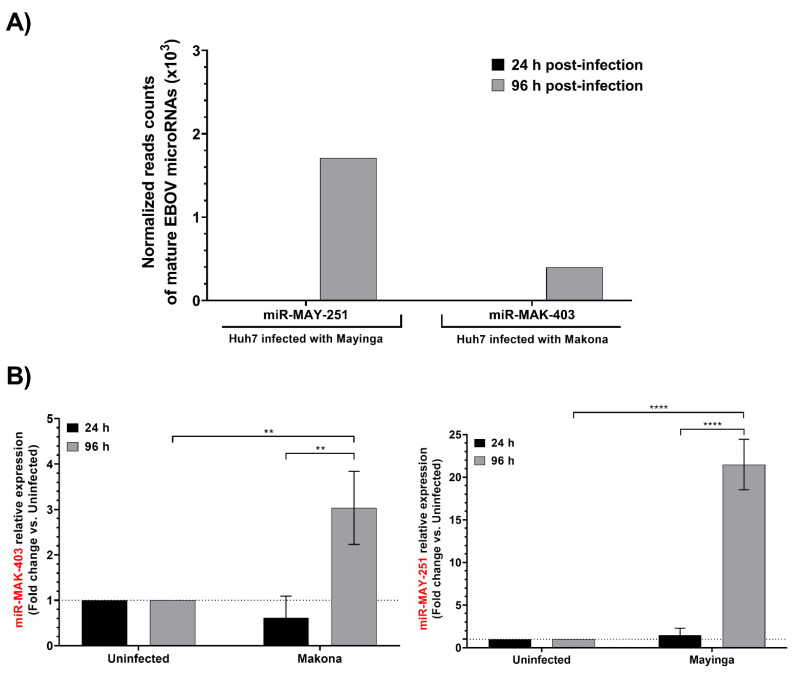
Expression of EBOV miR-MAY-251 and miR-MAK-403 in EBOV-infected Huh7 cells. (**A**) MicroRNA expression profiling data for the two putative EBOV microRNA sequences at 24 h and 96 h post-infection (see Section 4 and reference [33]). (**B**) RT-qPCR confirmation of the RNA-seq data showing relative expression of EBOV miR-MAY-251 and EBOV miR-MAK-403 in Huh7 cells infected for 24 h and 96 h with EBOV Mayinga or Makona variants, respectively. qPCR data were normalized with UniSp6 spike-in, reported to control (uninfected), and expressed with a relative quantitation method (ddCT). qPCR data were calculated from three biological replicates and expressed as means ± SD. Statistical analysis: The two-way analysis of variance (ANOVA) and the Sidak multiple comparisons test were used for statistical analysis. Statistically significant differences (fold change vs. uninfected) are indicated as follows: ** *p* < 0.01; **** *p* < 0.0001.

**Figure 4 ijms-23-05228-f004:**
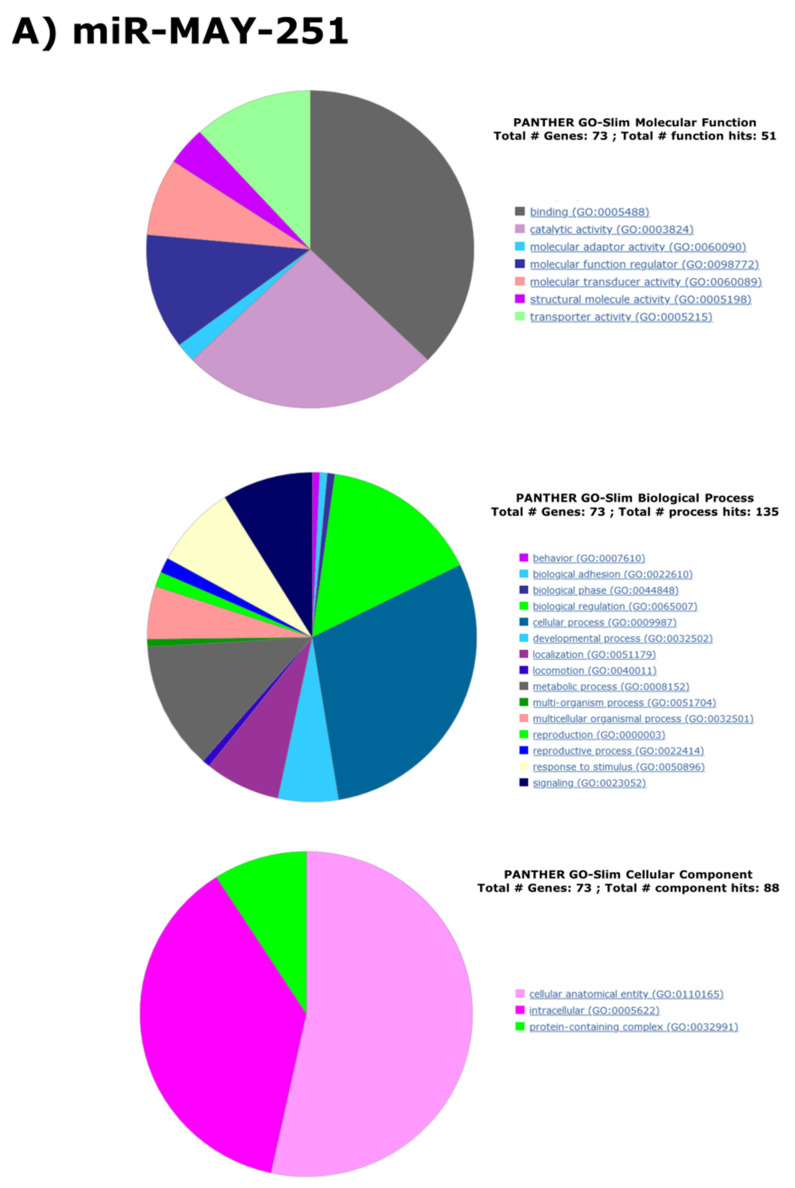
Pie charts of GO-Slim analysis based on EBOV miR-MAY-251 (**A**) and EBOV miR-MAK-403 (**B**) targets. EBOV miR-MAY-251 and EBOV miR-MAK-403 targets with scores >80% were subjected to the publicly available database PANTHER for gene function analysis. “Biological process”, “molecular function” and “cellular component” ontologies are represented here.

**Figure 5 ijms-23-05228-f005:**
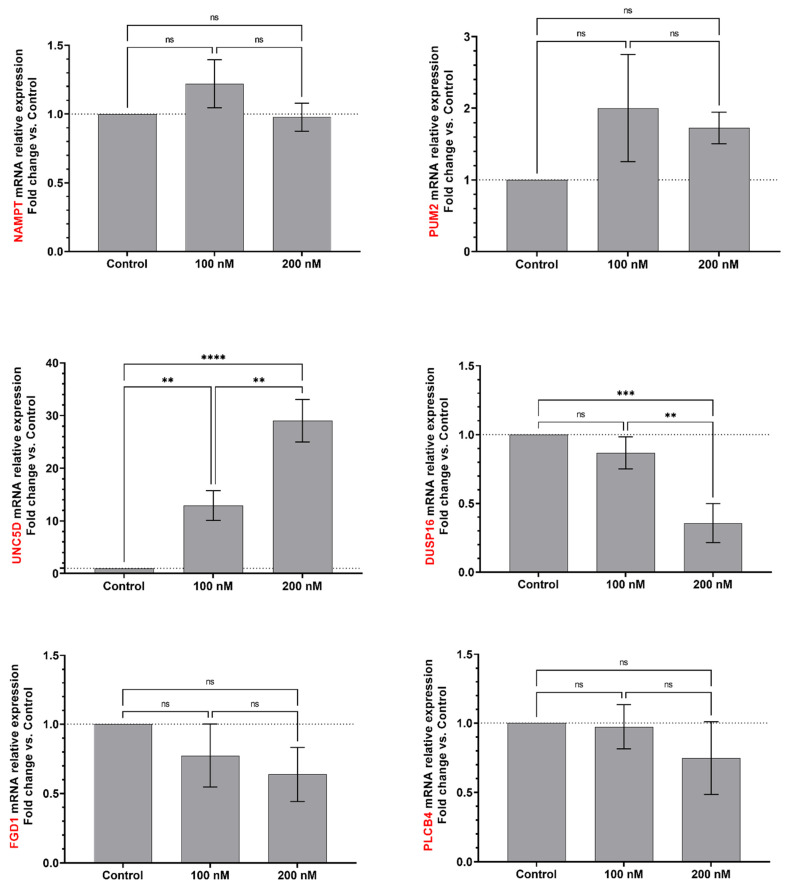
EBOV miR-MAY-251 may regulate several human host mRNA targets. NAMPT, PUM2, UNC5D, DUSP16, FDG1 and PLCB4 genes, which are potentially involved in viral infection mechanisms, were monitored by RT-qPCR (mRNA relative expression) after transfection of EBOV miR-MAY-251 mimic (100 and 200 nM) in Huh7 cells. qPCR data were normalized with a reference gene (ACTB), reported to control (mock control) and expressed with a relative quantitation method (ddCT). Statistical analysis: Data were calculated from three biological replicates and expressed as means ± SD. One-way analysis of variance (ANOVA) and Tukey’s multiple comparisons test were used, and statistically significant differences (fold change vs. control) are indicated as follows: ** *p* < 0.01; *** *p* < 0.001; **** *p* < 0.0001. ns, non significant.

**Figure 6 ijms-23-05228-f006:**
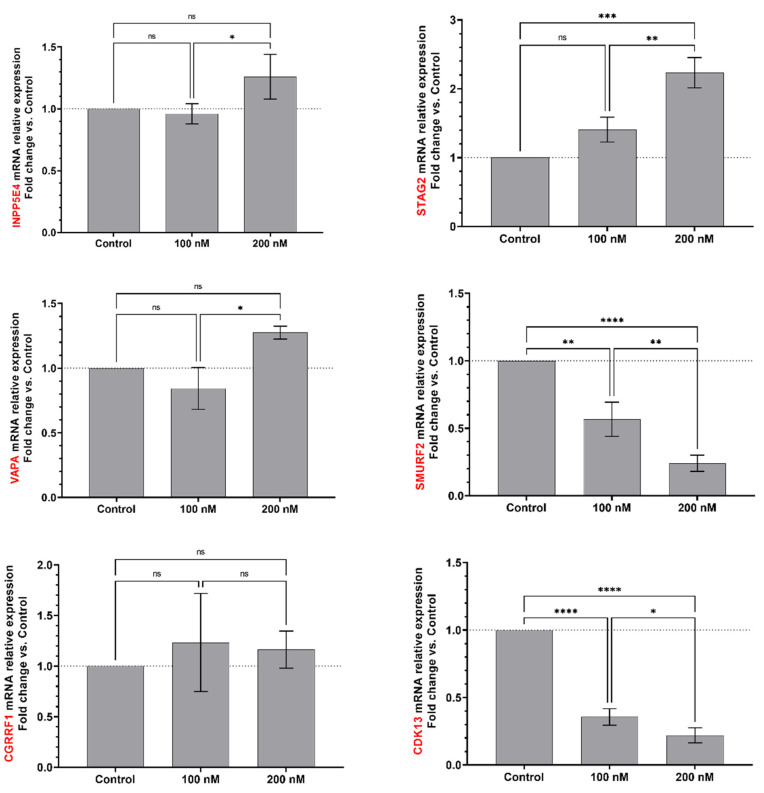
EBOV miR-MAK-403 may regulate several human host mRNA targets. INPPE5, STAG2, VAPA, SMURF2, CGRRF1, CDK13, WDR7 and P2YRY13 genes, which are potentially involved in viral infection mechanisms, were monitored by RT-qPCR (mRNA relative expression) after transfection of EBOV miR-MAK-403 mimic (100 and 200 nM) in Huh7 cells. qPCR data were normalized with a reference gene (ACTB), reported to control (mock control) and expressed with a relative quantitation method (ddCT). Statistical analysis: Data were calculated from three biological replicates and expressed as means ± SD. One-way analysis of variance (ANOVA) and Tukey’s multiple comparisons test were used, and statistically significant differences (fold change vs. control) are indicated as follows: * *p* < 0.05; ** *p* < 0.01; *** *p* < 0.001; **** *p* < 0.0001. ns, non significant.

**Figure 7 ijms-23-05228-f007:**
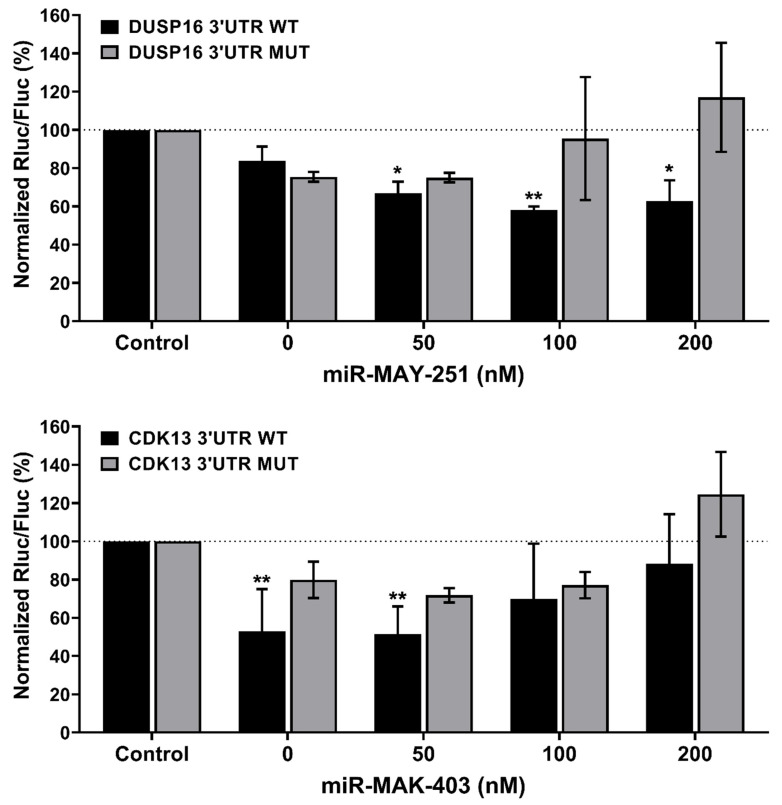
Modulation of human host mRNA target by EBOV microRNAs through their 3′ UTRs. Huh7 cells were co-transfected with EBOV miR-MAY-251 (**upper panel**) or miR-MAK-403 (**lower panel**) mimic (0, 50, 100, and 200 nM) and a psiCHECKII reporter construct (50 ng; see Appendix A), in which the Rluc reporter gene was coupled with wild-type (WT) or mutated (MUT) DUSP16 (**upper panel**) or CDK13 (**lower panel**) 3′ UTR. An unrelated, negative RNA control (Control) was used for normalization, in addition to the internal normalizer Fluc. “0 nM” corresponds to the transfection reagent-only control. Statistical analysis: Data were calculated from three biological replicates and expressed as means ± SD. The two-way analysis of variance (ANOVA) and Dunnett’s multiple comparisons test were used, and statistically significant differences (fold change vs. mock) are indicated as follows: * *p* < 0.05; ** *p* < 0.01.

## Data Availability

The data presented in this study are available on request from the corresponding author. The data are not publicly available due to the temporary unavailability of the miRBase repository system.

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
