# Peer review of "Ebola Virus Encodes Two microRNAs in Huh7-Infected Cells"

_ijms, 2022, doi:10.3390/ijms23095228_

Round 1
Reviewer 1 Report
Major comments:
1. there is not sufficient evidence in the current manuscript to support these two miRNAs are regulating the host cellular protein levels, although the provided experiments show the probabilities.
At least, assays of co-localization, miRNA binding to the regulation site et al., are needed to support the core conclusions in the manuscript.
2. in ‘Introduction’, EBOV is specifically for Ebola virus, herein, 'Ebola virus' should be filovirus or Ebolavirus when talking about its species
3. in ‘Introduction’, ‘Zaire Ebolavirus’ is out of date, please use Ebola virus (EBOV) only
4. in result 2.1, what's the difference between the stringent and less stringent criteria? Need to elucidate in the method section
5. in result 2.3, need to be careful to draw this conclusion ‘The above data provide an experimental validation of the regulation of host genes by EBOV miR-MAY-251 miRNA.’:
(1) not sufficient evidence to support direct regulation
(2) why UNCD5 level increased significantly? can't just say regulated by miR-MAY-251, and then say it's the target of miR-MAY-251
6. Compare the CDK13 results from Fig 7 and Fig 6, how the authors explain below difference:
(1) transcripts (100 and 200 nM groups) in fig 7 not significantly lower than control, while significant in fig 6
(2) in fig 7, not difference between 100 and 200, and 200 group even higher than 100, while 200 group shows significantly lower level than 100 in fig 6
Reviewer 2 Report
In their manuscript entitled, “Ebola virus encodes for two microRNAs in Huh7-infected cells,” Diallo et al. provide tantalizing evidence for the existence of two virus-derived miRNAs during EBOV infection of Huh7 cells. The authors identified the putative miRNAs, termed miR-MAY-251 and miR-MAK-403, from previous RNA-seq results obtained from Huh7 cells infected with either EBOV variant Mayinga or EBOV variant Makona. The authors demonstrate that expression of these putative miRNAs increases late during infection and they predict that the miRNAs target a number of pathways that could be involved in virus pathogenicity. Importantly, the authors are able to demonstrate that both miRNAs modulate the expression of host mRNA targets in transfected cells. While these data do not conclusively demonstrate the existence and functionality of these miRNAs, they do offer important preliminary evidence. Indeed, the authors do a good job of contextualizing their results in the Discussion section. Overall, this paper represents an interesting advancement in our understanding of miRNAs involved in filovirus infection. I have only minor concerns, detailed below.
- Filovirus nomenclature is used improperly throughout the manuscript. Several examples are listed below. The authors should revise their manuscript accordingly.
- Reston virus (RESTV) is repeatedly referred to as a strain of Ebola virus. This is not accurate. RESTV and Ebola virus (EBOV) are the type viruses of two distinct virus species (i.e., Reston ebolavirus and Zaire ebolavirus, respectively) that both belong to the Ebolavirus genus. The abstract, final paragraph of the Introduction, and beginning of the Discussion should be corrected.
- “Mayinga” and “Makona” can be more correctly defined as variants of EBOV not strains of EBOV.
- In the second paragraph of the Introduction, the authors state that “Ebola virus (EBOV) is a negative sense RNA virus consisting of six known species.” This is not correct. EBOV is the type virus of the species known as Zaire ebolavirus, which belongs to the Ebolavirus genus. The Ebolavirus genus is composed of six species.
- The authors should be careful not to use the abbreviation of a given virus for the virus species. For instance, Bundibugyo virus (BDBV) is the type virus of the species Bundibugyo ebolavirus. The species name (i.e., Bundibugyo ebolavirus) should not be abbreviated as BDBV.
- To avoid confusion, consider omitting the abbreviation “ZEBOV” all together.
- Species, genus, and family names should be italicized.
- Please use the virus abbreviations consistently throughout the manuscript. For example, “EBOV” is defined (correctly) as the abbreviation for “Ebola virus” early in the introduction, yet the terms “Ebola virus” and “Ebola” continue to occur repeatedly throughout the manuscript.
- Consider deleting the word “for” in the title of the manuscript.
- The first sentence of the Introduction is not correct. The 2013-2016 West African EBOV outbreak was not the “last” EBOV outbreak to occur in West Africa.
- Figure 2: It may be more accurate to replace the term “Transcription” with “Replication.” The figure depicts the negative sense RNA genome above the positive sense RNA genome, which serves as a replicative intermediate. Similarly, although the positive sense RNA genome is in the same sense as mRNA, it is not appropriate to label the genome as mRNA.
- In the Discussion section (page 16), the authors state that EBOV/Makona is newer and less well studied than EBOV/Mayinga and EBOV/Kikwit, which the authors suggest introduces a bias in data availability. While EBOV/Makona is newer, it may not be accurate to state that it is less well studied. The volume of data that came from patients infected with EBOV/Makona during the 2013-2016 West African outbreak far outweighs the data from all previous EBOV outbreaks combined. For sequence data in particular, we have a much wider and deeper understanding of EBOV/Makona than EBOV/Mayinga or EBOV/Kikwit.
- Can the authors compare the expression levels of the two miRNAs with the expression levels of NP and L transcripts in the infected Huh7 cells? This comparison could provide some additional context for interpreting miRNA expression and/or function.
- It is interesting that miR-MAK-403 differs by a single mismatch from the sequence in the 5’ UTR of L. Is this common for miRNAs? Is the sequence of miR-MAK-403 verifiably consistent? What mechanism might account for the sequence discrepancy?
- Can the authors assess the affect of the two miRNAs on the endogenous protein levels of DUSP16 or CDK13 by Western blot (or similar)? This would potentially help overcome the limitations of the reporter gene assay.
Round 2
Reviewer 1 Report
I appreciate the authors' efforts to address all those questions.
Although it is still insufficient to say the direct regulation relationship between these miRNAs and their potential targets, the authors provided sufficient explanation and discussion to support its novelty.
And good luck with your funding application. I understand it is sometimes a big pain.
Reviewer 2 Report
Thank you for addressing the majority of comments provided in the initial review report. The manuscript is improved.
At the risk of being too pedantic, I feel compelled to mention that there are still issues with filovirus nomenclature. Although the filovirus naming conventions are somewhat confusing and annoying, they are worth getting right. A more thorough explanation can be found here: PMID: 28653188.
The term "Ebolavirus" refers to a genus, which is a theoretical concept and not a virus. The term "Ebola virus (EBOV)" refers to the virus, which is a physical entity capable of infecting cells. The term "ebolavirus" (or "ebolaviruses") refers to all viruses that belong to the genus Ebolavirus, including Ebola virus. For instance, Ebolavirus (the genus) cannot infect cells, but Ebola virus and ebolaviruses can.
Thus, the second paragraph of the introduction could be more accurately re-written as follows:
"Ebola virus (EBOV) is a negative sense RNA virus that belongs to the Ebolavirus genus, which consists of six known species, including Zaire ebolavirus (to which EBOV belongs), Sudan ebolavirus, Bundibugyo ebolavirus, Tai forest ebolavirus, Reston ebolavirus, and Bombali ebolavirus."
Other instances of the use of the term "Ebolavirus" should be carefully reassessed where they occur throughout the manuscript. For example:
- "Indeed, the Ebolavirus genome codes for 9 proteins...": The genus Ebolavirus does not have a genome, but Ebola virus or the ebolaviruses do.
- "Several studies have detected the presence of Ebolavirus-derived miRNAs...": miRNAs cannot be derived from a genus; replace with "EBOV-derived miRNAs" or "ebolavirus-derived miRNAs"
- "...Huh7 infected with Ebolavirus...": cells cannot be infected by a genus; replace with "EBOV" or "ebolaviruses"
